# Single-Molecule Real-Time (SMRT) Full-Length RNA-Sequencing Reveals Novel and Distinct mRNA Isoforms in Human Bone Marrow Cell Subpopulations

**DOI:** 10.3390/genes10040253

**Published:** 2019-03-27

**Authors:** Anne Deslattes Mays, Marcel Schmidt, Garrett Graham, Elizabeth Tseng, Primo Baybayan, Robert Sebra, Miloslav Sanda, Jean-Baptiste Mazarati, Anna Riegel, Anton Wellstein

**Affiliations:** 1Department of Oncology, Lombardi Comprehensive Cancer Center, Georgetown University, Washington, DC 20007, USA; anne.deslattesmays@jax.org (A.D.M.); mos6@georgetown.edu (M.S.); garrett.graham@georgetown.edu (G.G.); ms2465@georgetown.edu (M.S.); jmazarati@gmail.com (J.-B.M.); ariege01@georgetown.edu (A.R.); 2The Jackson Laboratory, Farmington, CT 06032, USA; 3Pacific Biosciences, Menlo Park, CA 94025, USA; etseng@pacificbiosciences.com (E.T.); pbaybayan@pacificbiosciences.com (P.B.); 4Icahn School of Medicine at Mount Sinai, Institute for Genomics and Multi-scale Biology, New York, NY 10029, USA; robert.sebra@mssm.edu; 5Biomedical Center, National Reference Laboratory, Kigali, Rwanda

**Keywords:** full length RNAseq, mRNA isoforms, protein isoforms, bone marrow cell subpopulations

## Abstract

Hematopoietic cells are continuously replenished from progenitor cells that reside in the bone marrow. To evaluate molecular changes during this process, we analyzed the transcriptomes of freshly harvested human bone marrow progenitor (lineage-negative) and differentiated (lineage-positive) cells by single-molecule real-time (SMRT) full-length RNA-sequencing. This analysis revealed a ~5-fold higher number of transcript isoforms than previously detected and showed a distinct composition of individual transcript isoforms characteristic for bone marrow subpopulations. A detailed analysis of messenger RNA (mRNA) isoforms transcribed from the *ANXA1* and *EEF1A1* loci confirmed their distinct composition. The expression of proteins predicted from the transcriptome analysis was evaluated by mass spectrometry and validated previously unknown protein isoforms predicted e.g., for *EEF1A1*. These protein isoforms distinguished the lineage negative cell population from the lineage positive cell population. Finally, transcript isoforms expressed from paralogous gene loci (e.g., *CFD*, *GATA2*, *HLA-A*, *B*, and *C*) also distinguished cell subpopulations but were only detectable by full-length RNA sequencing. Thus, qualitatively distinct transcript isoforms from individual genomic loci separate bone marrow cell subpopulations indicating complex transcriptional regulation and protein isoform generation during hematopoiesis.

## 1. Introduction

Alternative splicing of pre-messenger RNA (mRNA) generates multiple transcript isoforms from a single gene that can code for proteins with distinct functions. Examples include distinct ligand recognition of the b- versus c-isoforms of fibroblast growth factor (FGF) receptors [1], the pro- versus anti-apoptotic activity of BCL-X [2], and FAS isoforms [3], or distinct hormone sensitivity due to long and short isoforms of a transcriptional co-activator, AIB1 [4] (reviewed in References [5,6,7]). Indeed, for >90% of human genes, an average of five transcript isoforms are predicted, suggesting a challenging complexity of the protein-coding transcriptome [8]. Unbiased methods such as RNA sequencing are crucial for unraveling the transcriptome complexity in different tissues or cells under diverse physiologic or pathologic conditions [9,10,11].

Bone marrow continuously replenishes differentiated blood cells from a small pool of stem cells. Studies of this process have provided many of the concepts of tissue regeneration from resident stem cells [12,13,14]. Here, we compared the transcriptomes of progenitor and differentiated bone marrow cell populations using single-molecule real-time (SMRT) full-length RNA-seq of unfragmented complementary DNAs (cDNAs) abbreviated here as “full length RNA-seq” (Pacific Biosciences, Menlo Park, CA, USA; Iso-Seq). For this, we isolated progenitor (lineage-negative) and differentiated (lineage-positive) cell populations from intact, freshly harvested human bone marrow tissues. We identified a multitude of novel transcript isoforms whose composition distinguished progenitor and differentiated cell populations at most of the single genomic loci interrogated. These differences in transcript composition were not uncovered by a de novo analysis of conventional, short-read RNA-seq of fragmented cDNAs that was run in parallel. We confirmed translation of transcripts by mass spectrometry of proteins extracted from the bone marrow and identified novel protein isoforms predicted from the transcript analysis. To our knowledge, this is the first study that performs full-length RNA sequencing on segregated hematopoietic cell subpopulations followed by mass spectrometry validation. We conclude that bone marrow cell subpopulations are distinguishable at the single gene level by qualitative differences in transcript isoform composition suggesting more complex transcriptome regulation during hematopoiesis than previously described.

## 2. Methods

*Healthy Bone Marrow Donors*—The study was reviewed and considered as “exempt” by the Institutional Review Board of Georgetown University (IRB # 2002-022). All methods were carried out in accordance with relevant guidelines and regulations. Freshly harvested bone marrow tissues were collected from discarded healthy human bone marrow collection filters that had been de-identified. cDNA libraries for the short-read RNA-seq and the full-length RNA-seq were isolated from two separate healthy donors. From one donor, three cDNA library preparations were generated for total bone marrow, lineage-negative, and lineage-positive cells. Full-length RNA-seq of the samples was done at Pacific Biosciences. From an additional donor, a cDNA library was generated from lineage-negative cells. This was sequenced at the Mt. Sinai Sequencing Facility. Independent sequencing results from the same donor and from the additional donor were used for validation of the full-length RNA-seq results (see Figure 1b).

*Healthy bone marrow cells*—mononuclear cells were isolated by Ficoll gradient centrifugation. In order to select for lineage-negative cells, bone marrow mononuclear cells were incubated with an antibody cocktail containing antibodies against CD2, CD3, CD5, CD11b, CD11C, CD14, CD16, CD19, CD24, CD61, CD66b, and Glycophorin A (Stemcell Technologies, Vancouver, British Columbia, Canada). Lineage-positive cells bound to the antibodies were removed by magnetic beads and lineage-negative cells obtained from the flow-through. To increase purity, lineage-negative cells were enriched two times.

*Short-read RNA-seq: Sequencing of fragmented cDNAs*—Total RNA was submitted to Otogenetics Corporation (Norcross, GA, USA) for RNA-seq. Briefly, the integrity and purity of total RNA were assessed using Agilent Bioanalyzer by OD260/280 ratio. Altogether, 5 μg of total RNA was subjected to rRNA depletion using the RiboZero Human/Mouse/Rat kit (Epicentre Biotechnologies, Madison, WI, USA). cDNA was generated from the depleted RNA using random hexamers or custom primers and Superscript III (Life Technologies, Carlsbad, CA, USA, catalog# 18080093). The resulting cDNA was purified and fragmented using a Covaris fragmentation kit (Covaris, Inc., Woburn, MA, USA), profiled using an Agilent Bioanalyzer, and Illumina libraries were prepared using NEBNext reagents (New England Biolabs, Ipswich, MA, USA). The quality, quantity, and the size distribution of the Illumina libraries were determined using an Agilent Bioanalyzer 2100. The libraries were then submitted for Illumina HiSeq2000 sequencing. Paired-end 90 or 100 nucleotide reads were generated and checked for data quality using FASTQC (Babraham Institute, Cambridge, UK), and DNAnexus (DNAnexus, Inc, Moutain View, CA, USA) was used on the platform provided by the Center for Biotechnology and Computational Biology (University of Maryland, Baltimore, MD, USA) [15]. A total of 159,043,023 non-strand-specific paired-end reads were collected from the total (T) and 35,126,712 strand-specific paired-end reads from the lineage-negative (N) bone marrow sample; 56.6% (T) and 51.3% (N) of the reads were mapped as a unique sequence using Tophat 2.

*Full-length RNA-seq*—total RNA was submitted to Pacific Biosciences (Menlo Park, CA, USA) or Icahn School of Medicine at Mount Sinai (New York, NY, USA). The integrity and purity of total RNA were assessed using Agilent Bioanalyzer and OD260/280 prior to submission. Full-length cDNA synthesis was done from polyA RNA using Clontech SMARTer PCR cDNA synthesis kit (Clontech Laboratories, Moutain View, CA, USA; [16]). Libraries were prepared after size selection of cDNAs into bins that contain 1–2 kb, 2–3 kb, and >3 kb cDNAs by the BluePippin size selection protocol (Sage Science, Beverly, MA, USA). These fractions were converted to single-molecule real-time (SMRT) libraries followed by SMRT sequencing. A total of 17 SMRT cells (7 cells 1–2 kb, 5 cells 2–3 kb, 5 cells 3–6 kb) were used to sequence the total bone marrow cell population, generating 234,078 single-molecule reads uniquely associated with non-redundant isoforms; 12 SMRT cells were used to sequence the lineage-negative population (5 cells 1–2 kb, 5 cells 2–3 kb, 2 cells 3–6 kb) generating 231,960 single-molecule reads uniquely associated with non-redundant isoforms Further, 8 SMRT Cells were used to sequence the lineage-negative population of another donor (4 cells <2 kb, 4 cells >2 kb) generating 195,614 single-molecule reads uniquely associated with non-redundant isoforms. Finally, 6 SMRT cells were used to sequence an alternative sample from the same donor, this was a lineage-positive population (2 cells 1–2 k, 2 cells 2–3 k, 2 cells 3–6 k) generating 74,434 single-molecule reads uniquely associated with non-redundant isoforms (see below, ToFU).

*Mass Spectrometry Analysis of Proteins by Nano LC-MS/MS*—proteins were extracted using 0.1% Rapigest (Waters Inc., Milford, MA, USA) in 25 mM ammonium bicarbonate Extracted proteins were reduced with 5 mM DTT for 60 min at 60 °C and alkylated with 15 mM iodoacetamide for 30 min in the dark. Trypsin (Promega, Madison, WI, USA) digestion (2.5 ng/μL) was carried out at 37 °C in Barocycler NEP2320 (Pressure BioSciences, Easton, MA, USA) for 1 h at 37 °C and then vacuum dried in Speed-vac (Labconco, Kansas City, MO, USA).

Tryptic peptides were analyzed on a NanoAcquity UPLC (Waters) by RP chromatography on a Symmetry C18 (3 µm, 180 µm, 20 mm) trap column and UPLC capillary column (BEH 300 Å, 1.7 µm, 150 mm × 0.75 µm) (Waters) interfaced with 5600 TripleTOF (AB Sciex, Framingham, MA, USA). Separation was achieved by a 250 min gradient elution with acetonitrile (ACN) containing 0.1% formic acid. The chromatographic method was composed of 5 min trapping step using 2% ACN at 15 μL/min and chromatographic separation at 0.4 μL/min as follows: Starting conditions 2% ACN; 1–180 min, 2%−60% ACN; 180−200 min, 60%−95% ACN; 200−220 min 95% ACN followed by equilibration 2% ACN for an additional 30 min. For all runs, 5 μL of sample were injected directly after enzymatic digestion. Analysis was conducted using an Information Dependent Acquisition (IDA) work flow with one full scan (400–1500 *m*/*z*) and 50 MS/MS fragmentations of major multiply charged precursor ions with rolling collision energy. Mass spectra were recorded in the MS range of 400–1500 *m*/*z* and MS/MS spectra in the range of 100−1800 *m*/*z* with resolution of 30,000 and mass accuracy up to 2 ppm using the following experimental parameters: Declustering potential, 80 V; curtain gas, 15; ion spray voltage, 2300 V; ion source gas 1, 20; interface heater, 180 °C; entrance potential, 10 V; collision exit potential, 11 V; exclusion time, 5 s; collision energy was set automatically according to *m*/*z* of the precursor (rolling collision energy). Data were processed using ProteinPilot 4.0 software (Sciex, Framingham, MA, USA) with a false discovery rate (FDR) of 1% [17,18]. Read-outs of the analysis are in Appendix A. For targeted measurements, an inclusion parent mass list was created according to in-silico tryptic digest of interesting sequences.

*Transcriptome Alignment and Assembly from Illumina Data*—Illumina reads were trimmed using Trimmomatic with the default parameters and the reads then aligned and assembled according to the Tuxedo suite protocol as described in Reference [15]. The genome of reference used was GRCh37 (hg19). The genes.gtf from this reference was used to guide the read alignment during the Tophat 2 step and Cufflinks2 [15]. Bowtie 2 indices were used for the genome reference. All computation was performed using Amazon Web Services and through the use of Starcluster software to manage the boxes. A Sun Grid Engine was employed to run the tasks. Reads were trimmed by Trimmomatic with the default parameters.

The specific qsub command for the total bone marrow (T) alignment are provided in the Supplemental Experimental Procedures.

*ToFU* (=**T**ranscript is**o**forms: **F**ull-length and **U**nassembled; also named Isoseq3)**—Reads obtained from the Pacific Biosciences RS II platform were run through the ToFU pipeline to obtain high-quality non-chimeric full-length reads [19]. The original python code wrapped several separate processes, permitting the software to be run on a high-performance computing cluster. This bioinformatic process begins with circular consensus reads and classifies the reads into full-length (5′ primer seen, polyA tail seen and 3′ primer seen) and non-full-length reads. Primers and the poly A/T tail sequences are then removed and the transcript strandedness determined. Consensus is used to correct random errors once transcripts are assembled into full-length similar clusters. Further consensus error correction is performed with the non-full-length transcripts using a function named Quiver to generate the final full-length error-corrected transcripts. These were collapsed to the longest transcript and their abundance information obtained. A master ID was created to permit the comparison of transcript isoforms obtained from one sample population to another. The abundance information was then converted to Transcripts per Million according to the specifications of Li et al. [20]. Custom python and R Scripts were used to perform the analysis. These are available upon request.

To correct for random sequencing errors the ToFU generates high-quality fasta sequences with high accuracy due to the circular consensus reads, which are the input to the ToFU algorithm and do not suffer from the raw read error rate—raw read errors on the Pacific Biosciences platform are corrected through consensus due do the barbell primers used to construct the libraries resulting in a single molecule being read about 10 times by the polymerase in a typical library, thus correcting random errors. ToFU performs an inter-well consensus that involves a multi-step process of classifying reads as full length or not by determining the presence of both the 5′ and the 3′ primers, polyA tail and if it is chimeric. The non-full-length reads are used for further confirmation. Here we used only high-quality reads (>99%) to create our final consensus transcripts.

*Conversion of FPKMs (=Fragments Per Kilobase of transcript per Million mapped reads) to* transcripts per million *(TPMs)*—the conversion formula used is:TPM = FPKM/(sum of the FPKM over all genes/transcripts) * 10^6^(1)

*Conversion of ToFU abundance to TPMs*—the conversion formula was generated according to Li et al. [20]. Abundance was obtained as output from ToFU and was translated to transcripts per million (TPM). Code is available upon request.

*Transcript quality assessment by MatchAnnot*—MatchAnnot is a python script that compares aligned full-length SMRT RNA-sequencing transcripts to existing annotation files. By starting with a transcript annotation file, MatchAnnot compares the full-length transcript and identifies a transcript within the genomic locus that provides the best match. Proceeding with this transcript, MatchAnnot then determines if there are skipped exons, alternative acceptor, and donor sites, etc. Comparisons are made using the transcript annotation file and the aligned input file (sorted SAM format). Start and end coordinate numbers are used to determine the details of the previously unknown transcript. Scores of 0 and 1 are considered poor matches. Scores of 2 or greater are considered as acceptable transcripts with viable alternative splicing annotation.

*Confirmation of novel transcript isoforms with blast*—to confirm novel transcripts isoforms, two BLAST-able databases were prepared separately for the total bone marrow (total.bm.non.ss) and lineage-negative short-read RNA sequences (lin.neg.ss) using the example script provided in Supplemental Experimental Procedures. For all reads where there was a gap, these were then compared with the full-length transcripts obtained from the alternative sample from the same donor (lineage-positive) and an alternative donor sample (lineage-negative) to arrive at the coverage numbers reported in Appendix A.

*Open Read Frame (ORF) Prediction*—ORFs were generated using the ANGEL software publically available at GitHub (San Francisco, CA, USA), and through the use of SerialCloner 2.6.1. Franck Perez [SerialBasics]. ORFs accepted were the first ORFs and not necessarily the largest. In some cases, both the first and the largest ORF were included and designated with letters a, b, etc., appended to the end of the name assigned.

*Multiple Sequence Alignment*—multiple sequence alignment was done using Clustal Omega available at EBI (European Bioinformatics Institute, Hinxton, UK).

*Sequence Alignment Editing*—the sequence alignments were edited using BioEdit version 7.2.5. This Sequence Alignment Editor written for the windows environment was run on OSX Yosemite on a Mac Book Pro through the use of the wine version 1.6.2, a windows emulator available for download, and installation through Home Brew version 0.9.5.

*Naming Convention*—the names generated by the cDNA Primer software were used to create the nomenclature that relates to the deposited transcript structures as well as the uniprot deposited protein isoforms.

*Quantitation of Abundance*—all figures were generated through custom R scripts permitting alignment of abundance with isoforms. The full-length sequence reads were aligned to the hg19 reference genome using gsnap. The fragmented sequence reads were aligned to the hg19 reference using Tophat 2 and the aligned and paired reads were assembled using Cufflinks 2 and genes.gtf also from hg19 annotation as a guide [15]. Quantitation was as reported by Cufflinks in FPKMs and these were transformed to TPMs according the above formula. Data structures using Bioconductor packages GRanges were used to unify the results. All figures generated in R were edited within Adobe Illustrator.

*Code availability*—custom code is available upon request.

## 3. Results

We compared the total human bone marrow cell population (T) dominated by differentiated cells with the small (<1%) subpopulation of lineage-negative progenitor cells (N) using single-molecule real-time (SMRT) sequencing of unfragmented cDNAs (abbreviated here as “full-length RNA-seq”; Figure 1; [19]). Samples were also analyzed at 20 and 100 million read depths by conventional RNA-seq, a method that relies on the computational assembly of transcripts from short-read sequences of fragmented cDNAs [15,21]. We first focus on the analysis of two representative genes that are abundantly expressed in hematopoietic cells (*EEF1A1* and *ANXA1*). Utilizing complete transcriptomes generated either by full-length or short-read RNA-seq, we then describe the composite results and provide a comparison with protein fragments identified by mass spectrometry. The overall experimental flow is shown in Figure 1.

*Distinct novel transcript isoforms of EEF1A1 detected in bone marrow cell subpopulations by full-length RNA-seq*—eukaryotic translation elongation factor 1 α 1 (*EEF1A1*) is a highly abundant, conserved protein that delivers aminoacyl-tRNAs to the ribosome during protein synthesis but has also been found to contribute to additional cellular functions [22]. The *EEF1A1* gene spans 5.2 kb on chromosome 6q13 and results in a 3.5 kb transcript that contains six protein coding exons (RefSeq HG19; Figure 2a). Short-read RNA-seq of total and lineage-negative bone marrow populations matched to the known reference transcripts translating for the canonical open reading frame (ST1-C, ST2-C*, in red for the Total; SN1-C, in blue for the lineage-negative; Figure 2b). One of the transcript isoforms, however, did not include the long 3′ UTR (ST2-C*) and one novel isoform in lineage-negative cells (SN2-3) skipped four coding exons due to an alternative splice acceptor and did not contain the long 3′ UTR in the RefSeq data base.

In contrast, the *EEF1A1* transcript isoforms obtained from full-length RNA-seq (Figure 2c) showed eight transcript isoforms each translating for the same canonical protein isoform (P68104; see Figure 3a). In total, seven of these isoforms were found in the lineage-negative (N4-C*, N9-C*, etc.,) and one was found in the total bone marrow cell population (T2-C*). An additional 34 previously unknown transcript isoforms were identified, 14 of which were found in the total and 28 in the lineage-negative cell populations. Exon-skipping was seen in 22 (e.g., T9-28), exon-splitting in 2 (T8-26, N13-10), and alternative donor/acceptor sites in 30 transcript isoforms (e.g., T7-25). No transcripts were found that contained the long 3′ UTR annotated in RefSeq (Figure 2c). The estimated abundance of transcript isoforms is shown in Figure 2d,e.

*A novel protein of EEF1A1 predicted from full length RNA-seq confirmed by mass spectrometry*—open reading frames predicted from the transcript isoforms of *EEF1A1* were aligned to the canonical protein and overlaid with its 3D structure (Figure 3a,c). Of the 21 novel protein isoforms predicted (Figure 3a,e), the N7 protein contained a unique tryptic peptide fragment that was distinct from the canonical EEF1A1 protein and thus potentially detectable by mass spectrometry. The N7 transcript is found only in the lineage-negative cell population at low abundance (21 TPM; see Figure 2e) and is predicted to code for a 205 aa protein that lacks the central 258 amino acids of the EEF1A1 protein joining Y86 with V344 (Figure 3a). A unique tryptic peptide fragment spanning Y86 to V344 was detected by mass spectrometry analysis of proteins extracted from lineage-negative cells (Figure 3b) and thus confirms that the N7-11 transcript is translated into a protein expressed at sufficient levels detectable by mass spectrometry. It is noteworthy that the protein segment containing the Y86-V344 junction also provides a distinct epitope that is not present in the canonical protein.

*Novel ANXA1 transcript isoform composition uncovered by full length RNA-seq distinguish bone marrow cell subpopulations*—annexins are known as organizers of membrane dynamics and include binding proteins for endocytosis, exocytosis, and other localization functions [24]. The *ANXA1* gene spans 18 kb on chromosome 9q21 with 12 coding exons and results in an approximately 1.5 kb transcript (Figure 4a). With the conventional short-read RNA-seq and computational transcript reconstruction, only the canonical *ANXA1* transcript was found in both total bone marrow and lineage-negative cell populations (Figure 4b). That is, there were no distinguishing transcript isoforms separating the total bone marrow from the lineage-negative cell population. In contrast, with full-length RNA-seq (Figure 4c), 38 transcript isoforms were identified; amongst these were transcript isoforms specific to the total bone marrow cell population (T7, T14) and transcript isoforms specific to the lineage-negative cell population (N2, N3, N4, N8). While there were shared transcript isoforms between the two cell populations, there were transcript isoforms found only in one population and not in the other—21 were found in the total and 17 were found in the lineage-negative cell population. Two of these novel isoforms predict the canonical protein, P04083 (T10-C*, N12-C*), while others contain distinct open reading frames generated by exon-skipping (25 isoforms; e.g., T16-12, N7-12), alternative donor and acceptor sites (11 isoforms; e.g., T19-6, N10-2), and intron retentions (9 isoforms; e.g., T6-14, N11-13).

*Proteins predicted from the ANXA1 transcript isoforms*—the canonical *ANXA1* protein contains four repeat regions (r1 to r4) of approximately 70 amino acids, each with a motif for calcium binding. These sequences are highlighted in yellow and the alignment of the predicted ORFs from transcript isoforms in total and lineage-negative bone marrow cells shows the overlaps with the canonical protein and with each other (Appendix A). Mass spec analysis of the proteins extracted from bone marrow confirmed the presence of matching peptides in both cell populations (red highlights, total; blue highlights, lineage-negative cells; Appendix A).

Four different transcript isoforms (T1-C, T10-C*, N1-C, and N12-C*) code for the canonical *ANXA1* protein. The T10-C* and N12-C* transcripts are structurally different from T1-C and N1-C, each containing a novel exon in the 5’ UTR (see Figure 4c). ORF 2 matches with the r4 repeat and is contained in seven different transcript isoforms whilst ORF 12, 13, and 14 match with the r1 to r3 repeats. These transcripts were detected in both cell populations. Additionally, ORF 3 (T14, T7), ORF 23 (N4, N8), and ORF 26 (N2, N3) are derived from two different transcript isoforms but found in only one of the cell populations (Appendix A). Finally, five novel protein isoforms due to exon-skipping are predicted from the transcript isoforms in total and lineage-negative cells (T13, T15, T19, N13, and N9). The identification of groups of isoforms seen only in one cell population but not the other shows that isoform composition can be sufficient to distinguish bone marrow cell subpopulations.

*High complexity of the transcriptome revealed by full-length RNA-seq*—as described above for *ANXA1* and *EEF1A1*, full-length RNA-seq of bone marrow cell populations revealed an up to 10-fold higher number of transcript isoforms than found by short-read RNA-seq. To investigate the universality of isoform complexity, we evaluated genes having <6 coding exons (*UBC*, *KLF6, LYZ*, *SAT1*) and found that RNA-seq detected 1 to 4 isoforms while full-length RNA-seq identified 12 to 36 isoforms. In a repeat sample, we identified as many as 15 isoforms with full-length RNA-seq, and in a sample from an additional donor we identified a maximum number of 48 isoforms for these genomic loci. We then considered loci with >6 coding exons. Short-read RNA-seq yielded a similar number of isoforms as it had with <6 exons, i.e., 1 to 3 isoforms compared with 31 to 43 transcript isoforms identified by full-length RNA-seq (Figure 5a). Similarly, in a repeat sample we identified 4 to 20 isoforms and from an additional donor 3 to 43 transcript isoforms at the genomic loci with >6 coding exons (EEF1A1, GLUL, HLA-E, CD74, PKM, and ANXA1).

We extended the analysis by arranging genes by mean exon number and identified the loci with the top five transcript isoform counts in each bin (Appendix A). Much to our surprise, the number of transcript isoforms identified by short-read RNA-seq plateaued at four isoforms irrespective of the number of exons in a given gene (Figure 5b). In contrast, full-length RNA-seq showed an increase in transcript isoform number with increasing complexity of genomic loci as indicated by the number of canonical exons (*p* = 0.0014). It is worth noting that an increase in sequencing depth for short-read RNA-seq from 20 to 100 million reads did not impact this maximum significantly (*p* > 0.05; Appendix A), an observation that matches with earlier predictions [25]. Thus, our analysis supports the notion that the complexity of the transcriptome will be underestimated by short-read RNA-seq regardless of the complexity of genomic loci evaluated.

*Matching of newly discovered and known transcript isoforms*—we compared the transcript isoforms found in the original runs as well as replicate samples from an additional donor with known transcript isoforms using matchAnnot (see Experimental Procedures); 26% to 58% of aligned reads showed the top matchAnnot score of >5 relative to known transcript isoforms deposited in the data base (Appendix A). Scores of ≥2, i.e., an acceptable match with some mismatched splicing, was seen for 88% to 98% of the transcripts detected. The qualitative differences in the transcript composition uncovered by full-length RNA-seq became apparent in this overall comparison where all transcript can be assigned to a genomic locus but only 39% of currently known transcript isoform compositions match with the isoforms described here (Appendix A). As shown above for a set of genes, the transcript isoform compositions enable one to distinguish lineage-positive from lineage-negative cell populations.

*Detection of novel transcripts of paralogous genes*—paralogous genes with conserved sequences pose a particular challenge to detection by short-read RNA-seq and we evaluated the potential of full-length RNA-seq to gain insights into genomic loci containing gene paralogs. Full-length RNA-seq identified transcripts for *CFD*, *GATA2*, *HLA-A*, *-B,* and *-C* that were missed by short-read RNA-seq, as shown in Figure 5c. The inability to detect or assign transcripts for these loci with short-read RNA-seq may be explained by the paralogous nature of the genes involved: *CFD* is located on chr 19 with *AZU1, PRTN3,* and *ELANE*. These four genes rank second in the list of regions of homozygosity cold spots on human autosomes and this genomic region underwent rapid Alu-mediated expansion during primate evolution creating the largest known microRNA gene cluster of the human genome [26]. Perhaps not surprisingly, transcript from a repetitive region of this kind was particularly problematic to resolve with short-read RNA-seq. Additionally, the ELANE and CFD proteins are 78% homologous (Appendix A; canonical proteins). Short-read RNA-seq uncovered transcripts from three of these four genes, missing *CFD*. That is not surprising, because blastn analysis matched *CFD* fragment sequences to *ELANE.* Using full-length RNA-seq, we were able to identify the transcripts as well as unique isoforms for each of the genes in these regions of homozygosity. Similarly, *HLA-A*, *HLA-B*, and *HLA-C* are paralogs with >80% identity that cannot be mapped appropriately and detected by short-read RNA-seq (Figure 5c and Appendix A). These data suggest that the presence of transcripts from paralogs adds to the complexity of alignments and obfuscate transcript reconstruction from short reads as well as the estimate of transcript abundance.

*Mass spectrometry identification of peptides predicted from the transcriptome analysis*—to assess the biological relevance of the transcript isoforms found and their predicted open reading frames, we identified detectable proteins in the bone marrow cell populations by mass spectrometry. We assessed the detection of matching peptides for a range of genomic loci that contain between 2 and 16 exons and were also associated with the highest number of transcript isoforms. Peptides were confirmed for 52 of the 150 transcripts depicted in Figure 5b (details in Appendix A).

As mentioned above, *HLA-A*, *-B,* and *-C* transcript isoforms were only detected by full-length RNA-seq (Figure 5c). It is noteworthy that mass spectrometry identified distinct peptides that were predicted for the *HLA-A*, *B,* and *C* transcript isoforms. The mass spec readings for three of the peptides are shown in Figure 5d–f and the alignment of these sequences in Appendix A. This supports the significance of the detection of these *HLA* transcripts and of the ORFs derived from the full-length RNA-seq.

*Information gained on transcript isoforms and their abundance*—for full-length RNA-seq, 56% of the transcripts mapped to loci with four or more exons and 31% mapped to loci with eight or more exons. In contrast, short-read RNA-seq mapped only 13% of transcripts to loci with >4 and 5% with >8 exons (Appendix A). Thus, full-length RNA-seq provides a significant (*p* < 0.0001) 4- to 6-fold gain in information. Given that over half of the detected transcripts are from multi-exon genes, the ability to span two exons with short reads may be inadequate to resolve a full-length transcript successfully without the addition of longer reads [21]. Also, ambiguous mapping of the short reads explains the high number of transcripts being mapped to genes with one or two exons (Appendix A). Our data also shows that short-read RNA-seq reaches a maximum of approximately four transcript isoform even for complex loci [8] whilst full-length RNA-seq shows a significant increase in transcript isoforms with increasing complexity of genomic loci (Figure 5b). As described in the more detailed analyses of *ANXA1* and *EEF1A1* above and an overview of genomic loci interrogated, full-length RNA-seq reveals complex posttranscriptional processing (Figure 6; Appendix A).

*Confirmation of novel isoforms*—we further confirmed the novel isoforms obtained in full-length RNA-seq in several different ways. First, we created a BLAST-able database using trimmed raw unaligned short-read RNA-seq data from each of the lineage-negative and total bone marrow populations. Limiting the results to only those reads with a 100 base pair coverage, novel transcript isoforms were confirmed by ensuring gapless coverage. Second, novel transcript isoforms were confirmed against full-length RNA sequencing libraries prepared from two additional samples (biological replicate samples). From a combination of these two approaches we confirmed an average of 83% of the transcript isoforms (range 74% to 100%; Appendix A). Third, we ran computational experiments with public data (Appendix A). From these analyses we found that the full-length RNA-seq mostly showed novel combinations of known junctions. 87% of all junctions for EEF1A1 have coverage from short-read data with 100% of the new transcript isoforms sharing an open reading frame with the known EEF1A1 protein. In total, 94% of all the junctions for ANXA1 have coverage from short-read data and 100% of the novel transcript isoforms share an open reading frame with known ANXA1 protein. Fourth, we assessed the transcription start site (TSS) matches with the 5′-end of the full-length RNA-seq. For EEF1A1, ~90% and for ANXA1, ~94% were confirmed, respectively (data and details in Appendix A). Fifth, we aligned the EEF1A1 and ANXA1 transcript isoforms described here with the EST database. We found that most of the isoforms showed 97% to 100% alignment identity with EST sequences with only a very few as low as 90% identity (Appendix A).

## 4. Discussion

Our transcriptome analysis of the human bone marrow progenitor and differentiated cell populations revealed a previously unknown complexity of the transcripts when analyzed by single-molecule real-time (SMRT) sequencing of unfragmented cDNAs, i.e., full-length RNA-seq. With a direct reading of full-length RNA sequences—rather than computational assembly from read fragments—open reading frames are predicted more reliably and a proteomics analysis for the respective predicted proteins showed that one third of them were present at high enough abundance to be detected by mass spectrometry. The full-length RNA-seq analysis described here is the most recent approach that relies on the long reads possible with the PacBio technology [27] and the use of full-length, non-fragmented cDNAs as templates. The template cDNAs are generated by the SMART (switching mechanism at 5′ end of RNA template) approach established earlier [16] and favors generation of full-length cDNAs that include the 5’-end of intact mRNA. This provides reliable assessments of open reading frames that start at the most 5’ translation initiation codon.

Numerous computational experiments were conducted exploring alternative methods, including using Trinity and Oasis to perform de novo assembly of the short reads alone, then combined with the full-length reads. These were experiments with varied success. Parts of transcripts would often be confirmed, but the process was both time consuming and computationally costly with unsatisfactory outcome. It is remarkable that the complexities of the transcriptome and hence distinction between subpopulations of bone marrow cells described above were confirmed but not identifiable de novo by the conventional, short-read RNA-seq of fragmented cDNAs. Indeed, short-read RNA-seq uses computational methods to reconstruct transcripts from the sequence fragments and can uncover novel exon-exon junctions. However, for acceptable performance, reconstruction requires a transcript isoform database to resolve the origin of isoforms. This will bias assignments of sequence fragments and limits discovery of isoforms of higher complexity [28]. Thus, with the sequence fragments provided by short-read RNA-seq it is difficult to uncover the composition of a complex transcriptome as is evident from the present study and was described before [29,30]. Greater sequencing depth is one potential remedy to consider and appears to improve the detection of complex transcripts only in *D. melanogaster* and *C. elegans* but not in *H. sapiens*. This is likely due to the lower number of transcript isoforms per genomic locus found in the invertebrates—fewer than 25% of genes in *C. elegans* and *D. melanogaster* give rise to more than two transcript isoforms, whereas human genes are currently annotated with an average of five transcript isoforms. Beyond this, short-read RNA-seq may identify too few isoforms due to ambiguous assignment of a given read to multiple loci. Altogether, this will impede the ability to detect novel isoforms [9,31,32].

The most striking examples that reveal the limitations of the short-read RNA-seq are paralogous gene loci where only full-length RNA-seq detected gene expression (Figure 5c). Mass spectrometry identified peptides in the bone marrow cell populations that match with the ORFs predicted from the full-length RNA-seq of the paralogous loci (Figure 5d–f and Appendix A). Interestingly, when the full-length transcript sequences were used as templates to search for matches in the raw data from short-read RNA-seq, 11% and 57% were confirmed for *HLA-A* and *-C*, respectively (Appendix A). These findings highlight the power as well as the limitations of the different approaches to RNA-seq.

An analysis of splicing complexity by gene size showed that the combination potential for exon splicing can increase exponentially [33]. The authors speculated that functional and evolutionary constraints are the reason why the number of transcripts thus far seen is less than the theoretical maximum. Our data indicate, however, that the number of transcripts reported to date may be limited by the short-read reconstruction approach. Full-length RNA-seq is able to reveal unique transcript isoform structures beyond those already discovered and allows for the unambiguous attribution of reads to defined genomic loci [11,30,33,34].

We conclude that single-molecule real-time full-length RNA-seq provides a map of the complexity of transcript isoform structures that eludes the transcriptome analysis by conventional short-read RNA-seq. In addition, open reading frames of proteins derived from these full-length transcript isoforms enables the comparison of the presence and absence of specific binding sites, domains, and regulatory regions in proteins that can be related to the cell population and physiologic or pathologic status [35]. In particular, the analysis of paralogous genes revealed the power of full-length RNA-seq to identify transcripts even for complex loci where short-read fragment sequencing failed. Transcriptome analysis by full-length RNA-seq provided a map of discrete transcript structures that can serve as qualitative markers for cell subpopulations in the bone marrow. This provides a qualitative expansion beyond the comparison of patterns based on the abundance of gene expression. Also, this indicates a more complex regulation of genes during hematopoiesis than previously appreciated.

Finally, as shown above, protein isoforms contain isoform-specific amino acid sequence junctions that represent epitopes specific for cell subpopulations in complex tissues. It is conceivable that pathologic regulation of splicing in diseases such as neurodegenerative disorders, metabolic syndrome, or cancer can generate neo-epitopes and elicit an immune response irrespective of genomic mutagenesis. Unraveling the complexity of the transcriptome complemented by analysis of resulting proteins thus may provide new diagnostic options and unique therapeutic targets.

## Figures and Tables

**Figure 1 genes-10-00253-f001:**
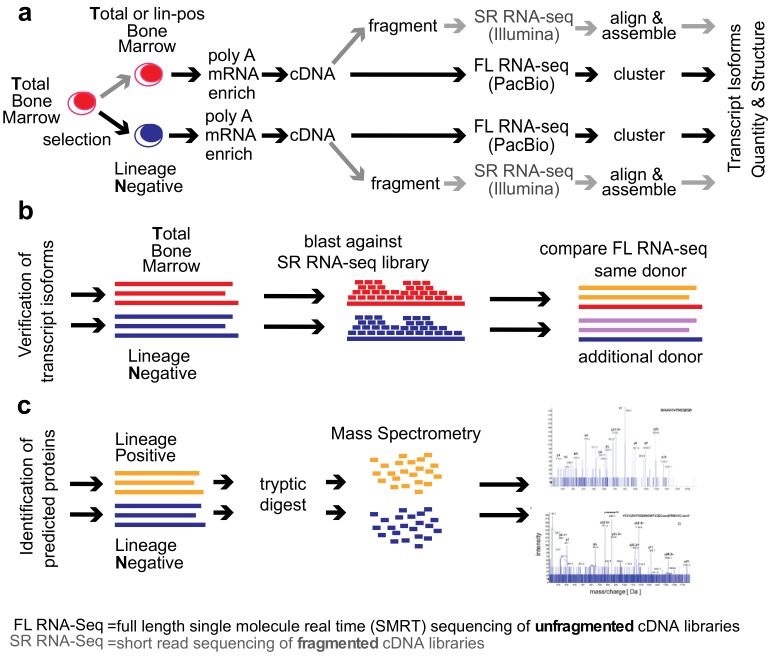
Graphical abstract of the Transcriptome analysis of human bone marrow (BM) cell populations. (**a**) Flow of experiments and analyses. Poly(A)^+^ RNA was isolated from Total (T, red) or lineage-negative BM cell populations (N, blue). Unfragmented, full-length complementary DNA (cDNA) libraries were subjected to single-molecule real-time (SMRT) RNA-seq (PacBio platform) or conventional short-read RNA-seq of fragmented cDNAs at 20 million (T) or 100 million (N) read depth (Illumina). Full-length RNA-seq data were processed using the ToFU platform. Illumina reads were first aligned and then assembled using the Tuxedo suite. The efficiency of the double selection for lineage-negative cells used here was confirmed by comparison of the abundance of standard markers of differentiated cells: CD14 = 6:1; CD16b = 25:1; CD24 = 109:1; CD45 = 11:1; CD66b = 16:1 expression ratio of lineage-positive to lineage-negative cells. (**b**) Validation of transcript isoforms identified by full-length RNA-seq. A BLAST-able library was generated from the short-read RNA-seq and used to align the isoforms identified by full-length RNA-seq. A separate full-length RNA-seq of an independent sample from the same donor, and a sample from a different donor were used for comparison. (**c**) Identification of proteins predicted by full-length RNA-seq using mass spectrometry of bone marrow cell extracts.

**Figure 2 genes-10-00253-f002:**
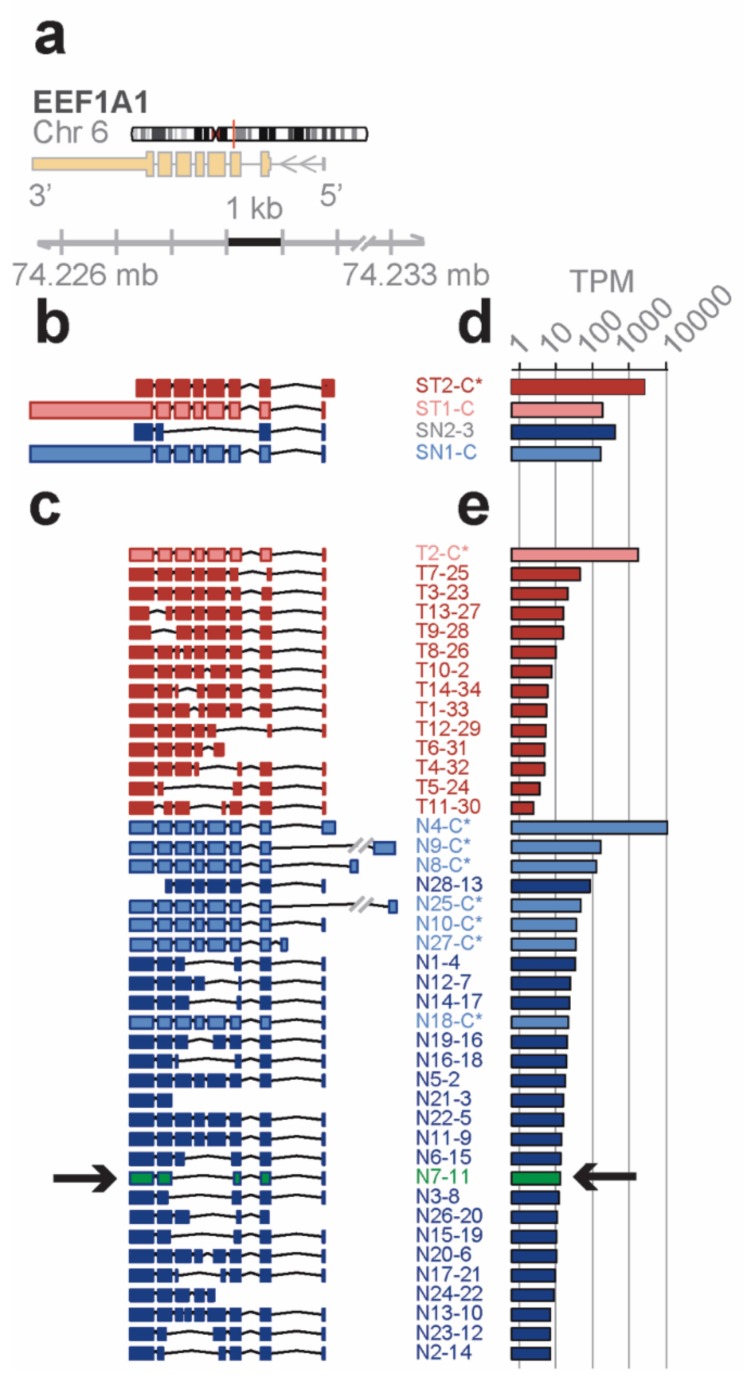
Messenger RNA (mRNA) isoforms of the *EEF1A1* gene. (**a**) Reference gene model from hg19. Arrows indicate direction of transcription. (**b**–**e**) Isoforms and abundances (TPM, transcripts per million reads) discovered in lineage-neg (blue) and total BM (red) cell populations by conventional short-read RNA-seq (**b**,**d**) or by full-length RNA-seq (**c**,**e**). Open reading frames (ORFs) and a novel protein isoform (arrows) that was confirmed by mass spectrometry for a unique peptide are shown in Figure 3. MatchAnnot confirmed the quality of the transcripts against gencode v19 (details in Methods). Abbreviations: S, short-read RNA-seq; C, canonical transcript and open reading frame (ORF); C*, non-canonical transcript isoforms with canonical ORF. ID#s of the isoforms are from the identifiers generated by the sequencing method. Ensembl IDs are included in Appendix A, NCBI Accession numbers in Appendix A.

**Figure 3 genes-10-00253-f003:**
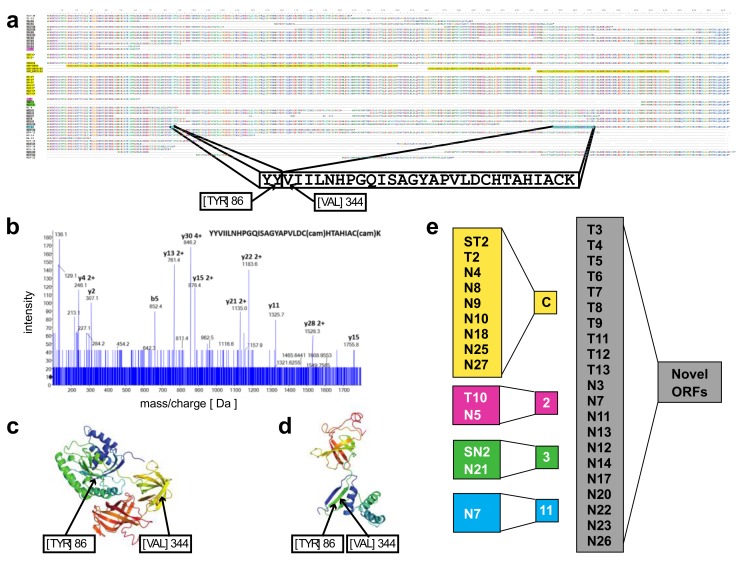
Proteins predicted from the transcript isoforms identified for *EEF1A1*. (**a**) Amino acid sequence alignments including the transcript isoform identifiers (see Figure 2b,c). The protein predicted from the N7-11 transcript isoform (highlighted in blue) joins Y86 and V344 in the canonical protein. The amino acid sequence of a predicted signature tryptic fragment peptide is shown. (**b**) Manually assigned fragmentation spectra from targeted analysis by Nano LC-MS/MS of proteins extracted from lineage-negative bone marrow cells with the sequence read of the N7-24 signature peptide; cam, carbamidomethylation. (**c**,**d**) Protein structure models were generated in Phyre2 [23]. The predicted structure of the canonical *EEF1A1* protein P68104 (c) and of the novel N7 protein (d) are shown. The c1g7ca template of P68104 used for the model covered 90% of the amino acids in the N7 isoform. (**e**) Higher magnification of transcript isoform identifiers that code for the canonical protein (yellow background). C, canonical RefSeq derived transcripts. C*, previously unknown transcript isoforms that code for the canonical protein. Transcript isoforms SN2-3 and N21-3 predict the same protein (green background).

**Figure 4 genes-10-00253-f004:**
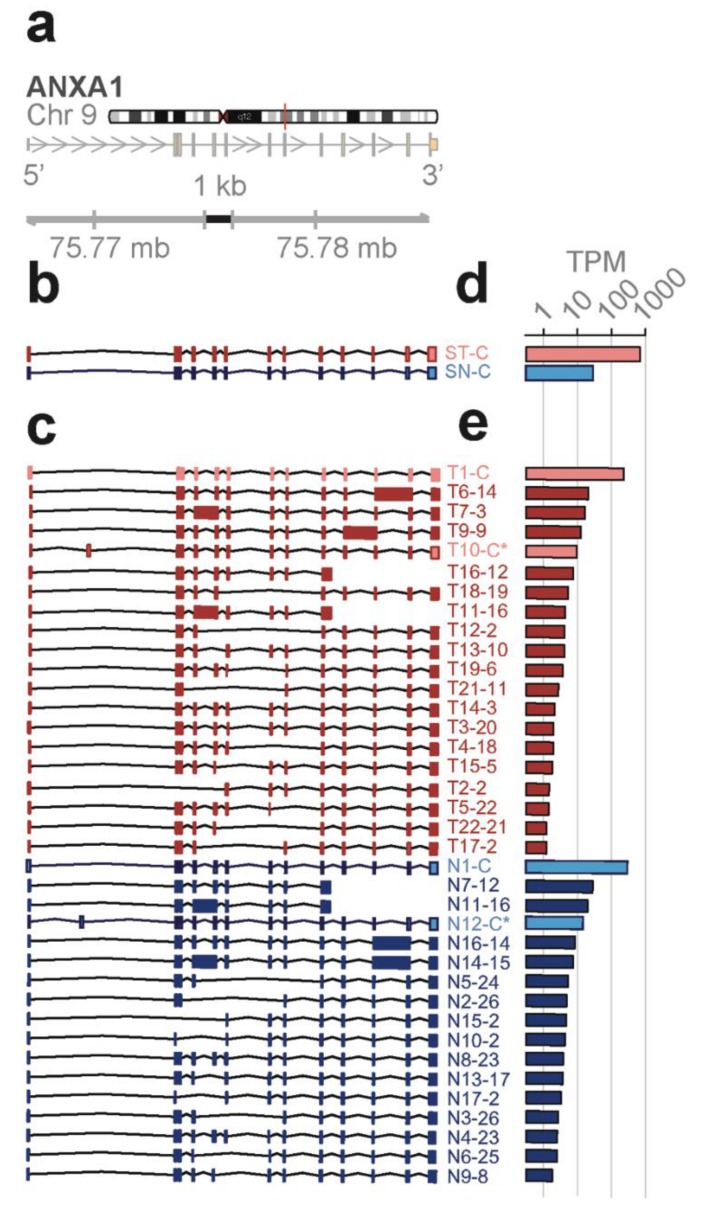
mRNA isoforms of the *ANXA1* gene. (**a**) Reference gene model from hg19. Arrows indicate direction of transcription. (**b**–**e**) Results for lineage-neg (blue) and total BM (red) from short-read RNA-seq (**b**,**d**) or full-length RNA-seq (**c**,**e**). Isoforms and abundances (TPM, transcripts per million reads) discovered in lineage-neg (blue) and total BM (red) population by short-read (**b**,**d**) or full-length RNA-seq (**c**,**e**). S, short-read RNA-seq; C, canonical transcript and open reading frame (ORF); C*, non-canonical transcript isoform with canonical ORF; ORFs are shown in Appendix A. ID#s of the isoforms are from the identifiers generated by the sequencing methods. Ensembl IDs are included in Appendix A, NCBI Accession numbers in Appendix A.

**Figure 5 genes-10-00253-f005:**
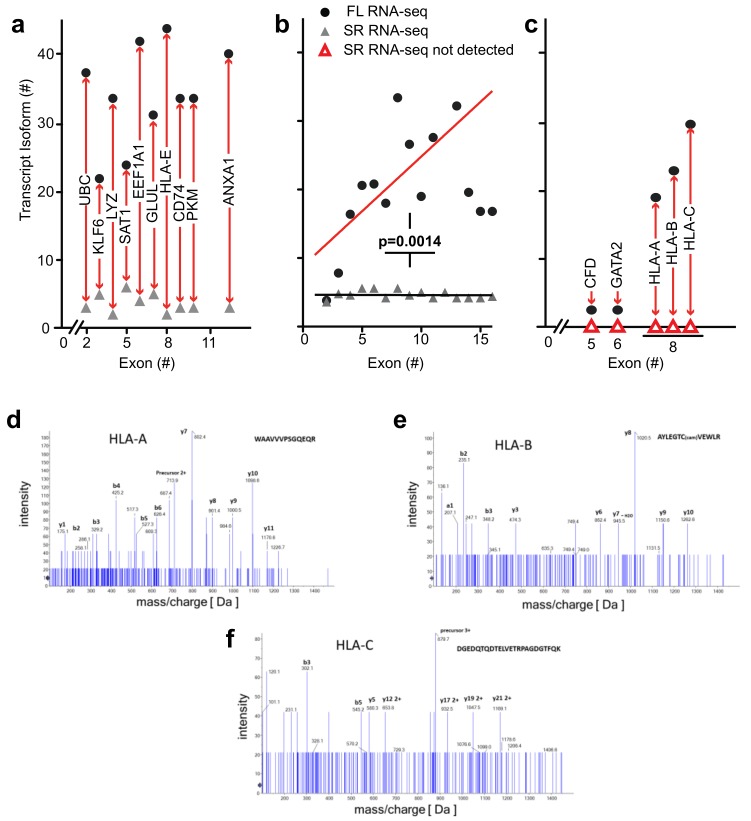
Transcript isoform detection for genomic loci of different complexity and redundancy. (**a**–**c**) The number of transcript isoforms detected (y-axes) by full-length (FL; filled circles) and short-read (SR; triangles) RNA-seq is shown relative to the number of canonical exons (x-axes; hg19 gene annotation). (**a**) Representative genes with 3 to 13 canonical exons. Full gene names and isoform numbers are provided in Appendix A. (**b**) Comparison of the mean of the transcript isoforms for the five most abundant genes with 2 to 16 exons from the analysis of all bone marrow cell populations (*p* = 0.0014; Chi-sq. for trend short-read (SR) RNA-seq versus full-length (FL) RNA-seq). The numbers of transcript isoforms for each of the subgroups are in Appendix A. (**c**) Transcripts and isoforms identified only by full-length RNA-seq. Gene names and numerical values are in Appendix A. (**d**–**f**) Manually assigned fragmentation spectra for peptides matching with HLA-A, -B, and -C transcripts. Peptides were detected using non-targeted shotgun proteomics analysis of proteins extracted from lineage-negative bone marrow cells. Detection of the peptides confirms expression of HLA-A, -B, and -C.

**Figure 6 genes-10-00253-f006:**
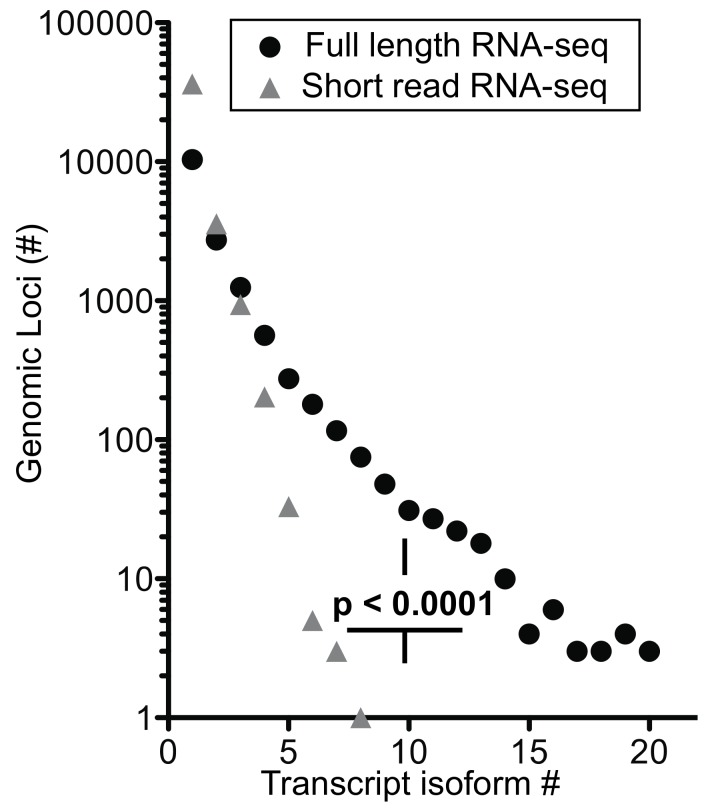
Number of transcript isoforms mapped to distinct genomic loci. Comparison of short-read (SR) versus full-length (FL) RNA-seq, *p* < 0.0001; Chi-square for trend. The distribution of exon counts per transcript isoform and transcript isoform counts per genomic locus are provided in Appendix A.

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
