# Peer review of "Single-Molecule Real-Time (SMRT) Full-Length RNA-Sequencing Reveals Novel and Distinct mRNA Isoforms in Human Bone Marrow Cell Subpopulations"

_genes, 2019, doi:10.3390/genes10040253_

Round 1

Reviewer 1 Report

The manuscript by Mays et al., describes the analysis of long-read (full length) RNA-seq from pools of bone marrow cells, including from lineage positive and lineage negative cells. They compare the short read to the long read RNA-seq data, with a particular focus on two genes EEF1A1 and ANXA1. They validate one of the unique transcripts of EEF1A1 using mass spectrometry. The authors observe some unique features of the long read RNA-seq, which suggests that short-read technology is very limited to detect novel transcript isoforms, independent of sequencing depth. They also demonstrate some of the advantages of tong read technology when considering transcripts that are very closely related (the HLAs). The manuscript is interesting, technically sound and well prepared. I have a few issues with the analysis that the authors should address: Major comments 1. The transcript abundances are currently presented as TPM on a log10 scale, but this drastically overestimates the relative abundances of the transcripts. These scales should be linear to illustrate to the reader clearly that the authors are discussing relatively minor isoforms. 2. It would be interesting to see if the short-read annotations support the novel long-read annotations if the long read annotations are given to Tophat/cufflinks. Does tophat cufflinks agree with the long read data if it is aware of the potential novel splice junctions? 3. I would like to see some genome pileups from the short-read data to see if the read density supports reads across the skipped and novel exons, particularly in the transcripts for EEF1A1 N9-C*, N25-C*, and ANXA1 T6-14, T7-3, T9-9. T11-16, N11-16, N16-14, N14-15. 4. EEF1A1 has a few novel transcription start sites. Have these been observed in any of the deepCAGE data? (It’s not an issue if it has not been seen, but would be helpful if it has been observed). 5.Similarly, EEF1A1 is one of the highest expressed genes in mammalian cells, and it seems very surprising that any novel exons or transcripts can be detected, considering the extensive EST databases. The authors should BLAST the EEF1A1 and ANXA1 transcripts against the EST databases, at least as an extra control. Minor comments 1. The manuscript appears to be missing an accession number for the deposited short read and long read RNA-seq data. 2. For Figure 2a and Fig 4a, all of the GENCODE transcripts should be shown for clarity. Looking at UCSC it seems a few of the long read transcripts correspond to known GENCODE isoforms, but it is unclear. The GENCODE isoforms should be labelled with their GENCODE accession number of clarity. 3. The commands for the analysis of the short-read RNA-seq is much appreciated in the supplement, as it makes it very clear what was done. It is however missing the equivalent commands for the long-read RNA-seq, which should be added for completion. 4.The paper is written in a style that seems to trust the long-read RNA-seq technology much more than the short-read technology, but whilst the long-read RNA-seq has great promise, the presence of some of these splice variants seem a little suspicious considering the very high expression levels of some of the these genes and their previous extensive study. The paper might be better written in a slightly more skeptical/cautious style, particularly if in future it turns out to have been some artifact in the long read RNA-seq that is currently unknown. 5. For the ANXA1 result, many of the transcripts look like retained intron events. The authors should comment on that possibility. 6.The novel EEF1A1 is referred to as N7-11 in Fig 2c,e and Fig 3e, but is labelled as N7-24 in Figure 3 figure legend, and as just ‘N7 transcript’ in the text (line 262). I am unclear if this is a mistake as I find the labelling system complex.

Author Response

Reviewer 1

The manuscript by Mays et al., describes the analysis of long-read (full length) RNA-seq from pools of bone marrow cells, including from lineage positive and lineage negative cells. They compare the short read to the long read RNA-seq data, with a particular focus on two genes EEF1A1 and ANXA1. They validate one of the unique transcripts of EEF1A1 using mass spectrometry. The authors observe some unique features of the long read RNA-seq, which suggests that short-read technology is very limited to detect novel transcript isoforms, independent of sequencing depth. They also demonstrate some of the advantages of tong read technology when considering transcripts that are very closely related (the HLAs). The manuscript is interesting, technically sound and well prepared. I have a few issues with the analysis that the authors should address:

Major comments

1. The transcript abundances are currently presented as TPM on a log10 scale, but this drastically overestimates the relative abundances of the transcripts. These scales should be linear to illustrate to the reader clearly that the authors are discussing relatively minor isoforms.

We drew gridlines in the TPM panels of Figures 2 and 4 to better visualize the abundance levels to the reader. The log scale allows to see the wide range much better than a linear scale would.

2. It would be interesting to see if the short-read annotations support the novel long-read annotations if the long read annotations are given to Tophat/cufflinks. Does tophat cufflinks agree with the long read data if it is aware of the potential novel splice junctions?

Yes, we have done that, see Table S6. We also ran additional computational experiments shown in Tables S10 and S11 with public data. This is described in the text:

“… Confirmation of novel isoforms - We further confirmed the novel isoforms obtained in full length RNA-seq in two different ways. First, we created a BLAST-able database using trimmed raw unaligned short read RNA-seq data from each of the lineage-negative and total bone marrow populations.  Limiting the results to only those reads with a 100 base pair coverage, novel transcript isoforms were confirmed by ensuring gapless coverage. Second, novel transcript isoforms were confirmed against full length RNA sequencing libraries prepared from two additional samples (biological replicate samples). From a combination of these two approaches we confirmed an average of 83% of the transcript isoforms (range 74% to 100%; Table S6). In addition, we ran computational experiments with public data (Tables S10 and S11). From the analyses we found that the full length RNA-seq mostly showed novel combinations of known junctions. 87% of all junctions for EEF1A1 have coverage from short-read data with 100% of the new transcript isoforms sharing an open reading frame with the known EEF1A1 protein. 94% of all the junctions for ANXA1 have coverage from short-read data and 100% of the novel transcript isoforms share an open reading frame with known ANXA1 protein. ….. ”

3. I would like to see some genome pileups from the short-read data to see if the read density supports reads across the skipped and novel exons, particularly in the transcripts for EEF1A1 N9-C*, N25-C*, and ANXA1 T6-14, T7-3, T9-9. T11-16, N11-16, N16-14, N14-15.

Supplemental Tables S10 and S11 hold the junction results from > 21K samples from the sequence read archive. We also ran additional computational experiments shown in Tables S10 and S11 with public data. This is described in the text:

“ .. Confirmation of novel isoforms - We further confirmed the novel isoforms obtained in full length RNA-seq in two different ways. First, we created a BLAST-able database using trimmed raw unaligned short read RNA-seq data from each of the lineage-negative and total bone marrow populations.  Limiting the results to only those reads with a 100 base pair coverage, novel transcript isoforms were confirmed by ensuring gapless coverage. Second, novel transcript isoforms were confirmed against full length RNA sequencing libraries prepared from two additional samples (biological replicate samples). From a combination of these two approaches we confirmed an average of 83% of the transcript isoforms (range 74% to 100%; Table S6). In addition, we ran computational experiments with public data (Tables S10 and S11). From the analyses we found that the full length RNA-seq mostly showed novel combinations of known junctions. 87% of all junctions for EEF1A1 have coverage from short-read data with 100% of the new transcript isoforms sharing an open reading frame with the known EEF1A1 protein. 94% of all the junctions for ANXA1 have coverage from short-read data and 100% of the novel transcript isoforms share an open reading frame with known ANXA1 protein. .. ”

4. EEF1A1 has a few novel transcription start sites. Have these been observed in any of the deepCAGE data? (It’s not an issue if it has not been seen, but would be helpful if it has been observed).

Tables S8 and S9 hold the deepCAGE data from the FANTOM5 (phase 1 and phase 2) project. In the case of EEF1A1, ~~90% of the TSS are confirmed.  For ANXA1, ~94% are validated. We write in the text:

We also assessed the transcription start site (TSS) matches with the 5’-end of the full length RNA-seq. For EEF1A1 ~90% and for ANXA1 ~94% were confirmed respectively (data and details in Tables S8 and S9).

5. Similarly, EEF1A1 is one of the highest expressed genes in mammalian cells, and it seems very surprising that any novel exons or transcripts can be detected, considering the extensive EST databases. The authors should BLAST the EEF1A1 and ANXA1 transcripts against the EST databases, at least as an extra control.

Supplemental Figures S5 and S6 hold the results from the blast against the EST databases and are described in the text:

“Finally, we aligned the EEF1A1 and ANXA1 transcript isoforms described with the EST data base and found that most showed 97% to 100% alignment with only a few as low as 90% identity (Figures S5 and S6).”

Minor comments

1.     The manuscript appears to be missing an accession number for the deposited short read and long read RNA-seq data.

The RNAseq data has been deposited and the NCBI numbers are shown in Supplemental Table S12 and included in the legend of Figure 2 and 4.

2.     For Figure 2a and Fig 4a, all of the GENCODE transcripts should be shown for clarity. Looking at UCSC it seems a few of the long read transcripts correspond to known GENCODE isoforms, but it is unclear. The GENCODE isoforms should be labelled with their GENCODE accession number of clarity.

Supplemental Tables S8 and S9 reference the gencode transcript identifiers to which the transcript isoforms sequenced have the best match and are included in the legends of Figure 2 and 4.

3.     The commands for the analysis of the short-read RNA-seq is much appreciated in the supplement, as it makes it very clear what was done. It is however missing the equivalent commands for the long-read RNA-seq, which should be added for completion.

Text regarding the specifics of running the ToFU are now included

4.     The paper is written in a style that seems to trust the long-read RNA-seq technology much more than the short-read technology, but whilst the long-read RNA-seq has great promise, the presence of some of these splice variants seem a little suspicious considering the very high expression levels of some of these genes and their previous extensive study. The paper might be better written in a slightly more skeptical/cautious style, particularly if in future it turns out to have been some artifact in the long read RNA-seq that is currently unknown.

We edited some of the sections according to the critique.

5.     For the ANXA1 result, many of the transcripts look like retained intron events. The authors should comment on that possibility.

This is commented on in the text   “…. and intron retentions (9 isoforms; e.g. T6-14, N11-13)… .”

6.     The novel EEF1A1 is referred to as N7-11 in Fig 2c,e and Fig 3e, but is labelled as N7-24 in Figure 3 figure legend, and as just ‘N7 transcript’ in the text (line 262). I am unclear if this is a mistake as I find the labelling system complex.

This was corrected

Reviewer 2 Report

The autors apply a novel long-read sequencing technique to analysis of transcript isoforms, and show substantial differences between the number of isoforms found by long-read versus short-read transcriptome sequencing. The study is innovative and of interest to readers. A few minor edits would help strengthen the results:

"ToFu". While this is a great name for a computational technique, please provide a more descriptive heading to the section

The short read analysis involves mapping reads to a reference transcriptome. I suspect that more transcript isoforms could be identified by de novo assembly of the short-read transcriptome. It would be valuable for the readers to conduct this analysis and compare results.

Would a combined approach produce better results? Can the short reads be mapped to the long read transcriptome assembly to improve sequence quality?

A known issue with PacBio data is low quality of base calls. Can the authors address this issue? 

The mass-spec data should be able to detect exons missing from splice isoforms, did the authors observe this in the data?

The mass spec data confirmed 52 out of 150 transcripts, which appears to be low. Can the authors explain this low validation rate?

Author Response

Reviewer 2

Comments and Suggestions for Authors

The autors apply a novel long-read sequencing technique to analysis of transcript isoforms, and show substantial differences between the number of isoforms found by long-read versus short-read transcriptome sequencing. The study is innovative and of interest to readers. A few minor edits would help strengthen the results:

"ToFu". While this is a great name for a computational technique, please provide a more descriptive heading to the section

Additional descriptions of ToFU were added to the method section.

The short read analysis involves mapping reads to a reference transcriptome. I suspect that more transcript isoforms could be identified by de novo assembly of the short-read transcriptome. It would be valuable for the readers to conduct this analysis and compare results.

We comment on this in the text:

“Numerous computational experiments were conducted exploring alternative methods, including using Trinity and Oasis to perform de novo assembly of the short reads alone, then combined with the full length reads. These were experiments with varied success. Parts of transcripts would often be confirmed, but the process was both time consuming and computationally costly with unsatisfactory outcome.”

Would a combined approach produce better results? Can the short reads be mapped to the long read transcriptome assembly to improve sequence quality?

Yes, we have used the combined approach, see Table S6. We also ran additional computational experiments shown in Tables S10 and S11 with public data. This is described in the text:

“… Confirmation of novel isoforms - We further confirmed the novel isoforms obtained in full length RNA-seq in two different ways. First, we created a BLAST-able database using trimmed raw unaligned short read RNA-seq data from each of the lineage-negative and total bone marrow populations.  Limiting the results to only those reads with a 100 base pair coverage, novel transcript isoforms were confirmed by ensuring gapless coverage. Second, novel transcript isoforms were confirmed against full length RNA sequencing libraries prepared from two additional samples (biological replicate samples). From a combination of these two approaches we confirmed an average of 83% of the transcript isoforms (range 74% to 100%; Table S6). In addition, we ran computational experiments with public data (Tables S10 and S11). From the analyses we found that the full length RNA-seq mostly showed novel combinations of known junctions. 87% of all junctions for EEF1A1 have coverage from short-read data with 100% of the new transcript isoforms sharing an open reading frame with the known EEF1A1 protein. 94% of all the junctions for ANXA1 have coverage from short-read data and 100% of the novel transcript isoforms share an open reading frame with known ANXA1 protein. ….. ”

A known issue with PacBio data is low quality of base calls. Can the authors address this issue? 

We comment on that in the methods section:

“The ToFU generates high quality fasta sequences with high accuracy due to the circular consensus reads, which are the input to the ToFU algorithm and do not suffer from the raw read error rate: Raw read errors on the Pacific Biosciences platform are corrected through consensus due do the barbell primers used to construct the libraries resulting in a single molecule being read about 10 times by the polymerase in a typical library thus correcting random errors. ToFU performs an inter-well consensus that involves a multi-step process of classifying reads as full length or not by determining the presence of both the 5’ and the 3’ primers, polyA tail and if it is chimeric. The non-full length reads are used for further confirmation. Here we used only high quality reads (> 99%) to create our final consensus transcripts.”

The mass-spec data should be able to detect exons missing from splice isoforms, did the authors observe this in the data?

We observed this in the example we show in Figure 3.  

The mass spec data confirmed 52 out of 150 transcripts, which appears to be low. Can the authors explain this low validation rate?

Protein abundance is a major limitation for mass spec discovery in contrast to the RNA analysis. Thus, it was within the numerous expected limitations of the depth of mass spec detection of that we were able to confirm translation of one third of transcripts by matching peptides.

Round 2

Reviewer 1 Report

No problems.